# Bactericidal and Virucidal Efficacies and Safety of Puriton®

**So-Hyeon Bok [1,†], Min-Hee Kim [2,†], Soon-Young Lee [1], Chun-Sik Bae [3], Min-Jae Lee [4], Kwang-Ho Kim [5,*] and Dae-Hun Park [1,*]**

1   College of Korean Medicine, Dongshin University, Naju, Jeonnam 58245, Korea; bok_23@naver.com (S.-H.B.); asy390@naver.com (S.-Y.L.)
2   Department of Forestry, Chonnam National University, Gwangju 61186, Korea; minhee3947@naver.com
3   College of Veterinary Medicine, Chonnam National University, Gwangju 61186, Korea; csbae0313@hanmail.net
4   Department of Veterinary Medicine, Kangwon National University, Gangwon 24341, Korea; mjlee@kangwon.ac.kr
5   Kadesh, Inc., Garden Grove, CA 92841, USA
*   Correspondence: david3188051@gmail.com (K.-H.K.); dhj1221@hanmail.net (D.-H.P.); Tel.: +1-714-620-8866 (K.-H.K.); +82-10-9930-5494 (D.-H.P.)
†   These authors equally contributed.

**Abstract:** In 2016, infectious microbes were one of the leading causes of death, especially in developing countries. Puriton® is a mineral mixture consisting of biotite, kaolinite, montmorillonite, serpentine, clinochlore, and vermiculite, and evaluated antimicrobial activity in vitro and safety in vivo. Nine pathogens and opportunistic bacteria, namely *Salmonella typhimurium*, *Escherichia coli*, *Pseudomonas aeruginosa*, *Alcaligenes faecalis*, *Staphylococcus aureus*, *Enterococcus faecalis*, *Micrococcus luteus*, *Mycobacterium smegmatis*, and *Bacillus subtilis*, and the two viruses Zika and Influenza A/Duck/MN/1525/81 were used. A 26-week oral repeated safety study of Puriton® was conducted. Puriton® suppressed the bacterial proliferation, with a minimum proliferative rate of 91.1% in *B. subtilis* ATCC6633. The virucidal efficacy of Puriton® against Zika virus after 4 h and 18 h of contact time was significant in all groups treated with Puriton®. Twenty-six-week repeated oral administration of Puriton® was considered safe based on physiological results, such as behavior and blood cells/chemistry, and histopathological changes in the brain, heart, kidney, liver, and lung. We concluded that Puriton® exerted bactericidal and virucidal efficacies and was safe for 26-week repeated oral administration.

**Keywords:** Puriton®; mineral mixture; safety; bactericidal; virucidal

## 1. Introduction

In May 2018, the World Health Organization reported the top 10 causes of death in 2016 [1]. Although there was a difference in economic status between developed countries and developing countries, divided as low-income countries, lower-middle-income countries, upper-middle-income countries and high-income countries, the common death causes worldwide were bacterial infections. There were several deaths causes which were directly originated by microorganisms such as malaria, tuberculosis, and diarrheal diseases. In 2012, 2195 children died every day due to infected diarrhea, more than AIDS, malaria, and measles combined [2].

Bacterial infection results from the pathogenicity of bacteria. *Salmonella typhimurium* [3] and *Escherichia coli* [4] incur foodborne outbreaks, whereas *Pseudomonas aeruginosa* is related to many

opportunistic infectious diseases in various organs [5]. *Alcaligenes faecalis,* a resident flora in the gastrointestinal tract, and may cause opportunistic infections, such as chronic suppurative otitis media, peritonitis, and skin and soft tissue infection [6–8]. *Staphylococcus aureus*, a normal flora in the upper respiratory tract and skin/gut [9,10], can be pathogenic in morbid environments, causing skin infection, food poisoning, and bacteremia [11,12]. *Enterococcus faecalis* is found in healthy people, but can cause various infectious diseases, such as endocarditis, urinary traction infection, meningitis, and septicemia [13,14]. *Micrococcus luteus* exists not only on mammalian skin but also in the environment, such as in the soil, dust, water, and air; moreover, this species can cause septic shock in various disease-suffering people [15]. *Mycobacterium smegmatis* has been known as a non-pathogenic bacterium, but in some cases, it causes tuberculosis [16]. *Bacillus subtilis* is known to produce amylosin, a heat stable toxin, and *B. subtilis* contamination may cause an outbreak of food poisoning [17]. Microbial infections also cause an increase in metabolic disorders, such as diabetes, a trend which could be observed worldwide over last several years [18]. Defects of insulin modulation cause diabetes mellitus, and induce hyperglycemia; chronic hyperglycemia can lead to severe health problems [19].

Minerals are important elements composing organisms: they function as special constituents, such as heme in hemoglobin, co-enzymes, and regulators of the heartbeat, and they also maintain fluid balance and function in muscle contraction. Besides minerals in the body, many minerals have been used for their biological efficacies. Edible clays have been used as traditional remedies and food materials worldwide [20]. Clay can control the toxic effects of some materials such as metal and organic pollutants, and can discharge consumed toxins [21]. The biological functions of many minerals have been reported. For example, minerals play a role in the intestinal IgA production of deep-sea water organisms [22]. Natural spa minerals are known to exert analgesic effects [23]. Mineral block supplementation showed antioxidant effect and immunity increment in animals [24]. Puriton® is a mineral mixture in sterilized water at a pH of almost 7.0, is a transparent liquid type, and is used as an oral treatment.

In this study, we evaluated Puriton®'s biological efficacies, such as its antibacterial effect antiviral effect, and simultaneously its safety of repeated oral use.

## 2. Materials and Methods

### 2.1. Puriton® Components

Puriton® (Kadesh, Inc., Garden Grove, CA, USA) consisted of 20% biotite, 17% kaolinite, 16% montmorillonite, 12% serpentine, 8 mica, 4% clinochlore, 4% vermiculite, 3% muscovite, 2% brucite, 2% limestone, 2% illite, 1% zeolite, 1% orthoclase, and 8% other minerals in sterilized water (almost pH 7.0).

### 2.2. Bacterial Culture, Culturable Conditions, Virus Culture, and Cytopathic Effect (CPE) Examination

In this study, nine pathogens and opportunistic bacteria were used. *Alacligenes faecalis* ATCC1004, *Pseudomonas aeruginosa* ATCC BAA-1744, *Enterococcus faecalis* ATCC29212, *Micrococcus luteus* ATCC9341, and *Bacillus subtilis* ATCC6633 were obtained from American Type Culture Collection (ATCC, Manassas, VA, U.S.A.). We purchased 3 bacteria, *Escherichia coli* KCTC1923, *Salmonella typhimurium* KCTC1925, and *Staphylococcus aureus* KCTC1928, from the Korean Collection for Type Culture (KCTC, Jeonbuk, South Korea). Briefly, test organisms were mixed with distilled water or Puriton® ($3.4 \times 10^5$~$2.2 \times 10^6$ cfu/mL) which was made as liquid type under almost pH 7.0 using sterilized water for 30 min, and 10 μL mixture was loaded on to nutrient agar plates and incubated at 37 °C for 24 h. After incubation, the number of bacterial colonies was counted.

Zika virus (MR766 Uganda) and Influenza A/Duck/MN/1525/81 (H5N1) virus were used in this experiment. Each virus was treated with Puriton® (1 mL) in tubes in triplicate for each time point, 4 h or 18 h. A negative control (water) and positive control (70% EtOH) were included in each replicate. One set of toxicity control tubes was prepared in the same manner without a virus. Viruses were added

to the tubes for each time point (4 h and 18 h). Zika virus stock (10 μL) and H5N1 virus stock (100 μL) were added to the respective tubes and mixed thoroughly. The H5N1 virus stock had a lower titer; therefore, its volume was increased so that the highest concentration of drug tested was 90% after the addition of virus, whereas that for Zika virus was 99%. The tubes were incubated at 23 ± 3 °C for 4 h or 18 h. Following incubation, the samples were added to cell culture media at a 1:10 dilution, and serial log dilutions were performed. Diluted samples were each added to four wells of a 96-well plate, which contained 80–90% confluent *Canis familiaris* kidney-originated MDCK cells (NBL-2, ATCC® CCL-34™) for influenza virus and Vero 76 cells for Zika virus. Toxicity controls were diluted and plated in the same manner as described above. Half of the wells containing uninfected controls were spiked with a virus (30 $CCID_{50}$/well) to monitor antiviral activity (neutralization controls). The plates were incubated at 37 ± 2 °C and 5% $CO_2$. The cultures were scored for the presence or absence of CPE on day 3 for H5N1 and day 6 for Zika virus. The Reed–Muench method was used to determine end-point titers (50% cell culture infectious dose, $CCID_{50}$) of the samples [25]. The log reduction value (LRV) of Puriton® compared to the negative control (water) was calculated.

### 2.3. Animal Experiment

To evaluate the safety of Puriton®, male ICR mice were purchased from SamTako Bio Korea (Osan, Korea) and acclimatized to experimental conditions for 7 days. The mice were divided into three treatment groups and treated for 26 weeks with ad libitum access to the corresponding diet and water. The treatment groups were as follows: (1) control; (2) 400 μL Puriton®; and (3) 800 μL Puriton®. All experiments were approved by the Institutional Animal Care and Use Committee at Chonnam National University (Approval No. CNU IACUC-YB-2018-88).

### 2.4. Macroscopic, Physiological, and Histopathological Evaluations

All animals were observed twice per day for physiological changes, such as changes in movement, fur color/grooming, appetite, aggression, and body weight. Dietary consumption was measured once per week. At 3 h before euthanasia, feeding was restricted and the mice were anesthetized with intraperitoneal injections of 50 mg/kg Zoletil (Virbac, Fort Worth, TX, USA). After euthanasia, all organs were observed with the naked eye to identify any morphological changes due to toxicity. Whole blood was collected via the heart, and blood chemistry (total cholesterol, high-density lipid (HDL), low-density lipid (LDL), and glucose levels) analysis was conducted using a Hemavet Multispecies Hematology System (Drew Scientific Inc., Waterbury, CT, USA) and a HUJI Dri-Chem NX500i (FUJIFILM Corporation, Tokyo, Japan) (n = 8 per group). The brain, heart, kidney, liver, and lung were collected for the evaluation of any morphological changes due to Puriton®. All organs were fixed in 10% (*v/v*) formaldehyde solution, dehydrated in a graded ethanol series (99.9%, 90%, 80%, and 70%), and embedded in paraffin. The paraffin-embedded tissues were then sectioned (5 μm) longitudinally and stained with hematoxylin and eosin.

### 2.5. Statistical Analysis

The data are shown as the mean ± standard deviation (SD). Group differences were evaluated by one-way analysis of variance (ANOVA) followed by Dunnett's multiple comparison test. Statistical significance was set at $p < 0.01$.

## 3. Results

### 3.1. Puriton® Controlled the Colonies of Nine Pathogenic Bacteria

Puriton® completely eliminated four bacteria: *S. typhimurium* KCTC1925, *E. coli* KCTC1923, *P. aeruginosa* ATCC BAA-1744, and *A. faecalis* ATCC1004 at 30 min after treatment, and it strongly inhibited the growth of *S. aureus* KCTC1928, *E. faecalis* ATCC29212, *M. luteus* ATCC9341, *M. smegmatis* ATCC9341, and *B. subtilis* ATCC6633 by 99.9%, 97.8%, 96.4%, 93.9%, and 91.1%, respectively (Table 1).

Before the colonies were counted, photographs of the bacterial colonies were taken to allow qualitative observation of the bactericidal effect of Puriton® (Figure 1). Although the death rate of each bacterium was different, all of them (nine bacteria strains) were mostly controlled by Puriton®, with *B. subtilis* ATCC6633 showing the lowest rate of 91.1% (Figure 1E).

**Table 1.** Inhibition rate of Puriton® against nine pathogenic bacteria. This experiment was conducted once on a single day.

| NO | Micro-Organisms | Before Treatment CFU/mL | At 30 min after Treatment CFU/mL | Inhibition Rate (%) |
|----|-----------------|-------------------------|----------------------------------|---------------------|
| 1 | *Alacligenes faecalis* ATCC1004 | 21,750,000 | 0 | 100.0 |
| 2 | *Escherichia coli* KCTC1923 | 317,200 | 0 | 100.0 |
| 3 | *Pseudomonas aeruginosa* ATCC BAA-1744 | 1,147,200 | 0 | 100.0 |
| 4 | *Salmonella typhimrium* KCTC1925 | 170,000 | 0 | 100.0 |
| 5 | *Staphylococcus aureus* KCTC1928 | 173,400 | 700 | 99.9 |
| 6 | *Enterococcus faecalis* ATCC29212 | 13,700,000 | 40,000 | 97.8 |
| 7 | *Micrococcus luteus* ATCC9341 | 23,650,000 | 850,000 | 96.4 |
| 8 | *Mycrobacterium smegmatis* ATCC9341 | 8,400,000 | 90,000 | 93.9 |
| 9 | *Bacillus subtilis* ATCC6633 | 3,100,000 | 275,000 | 91.9 |

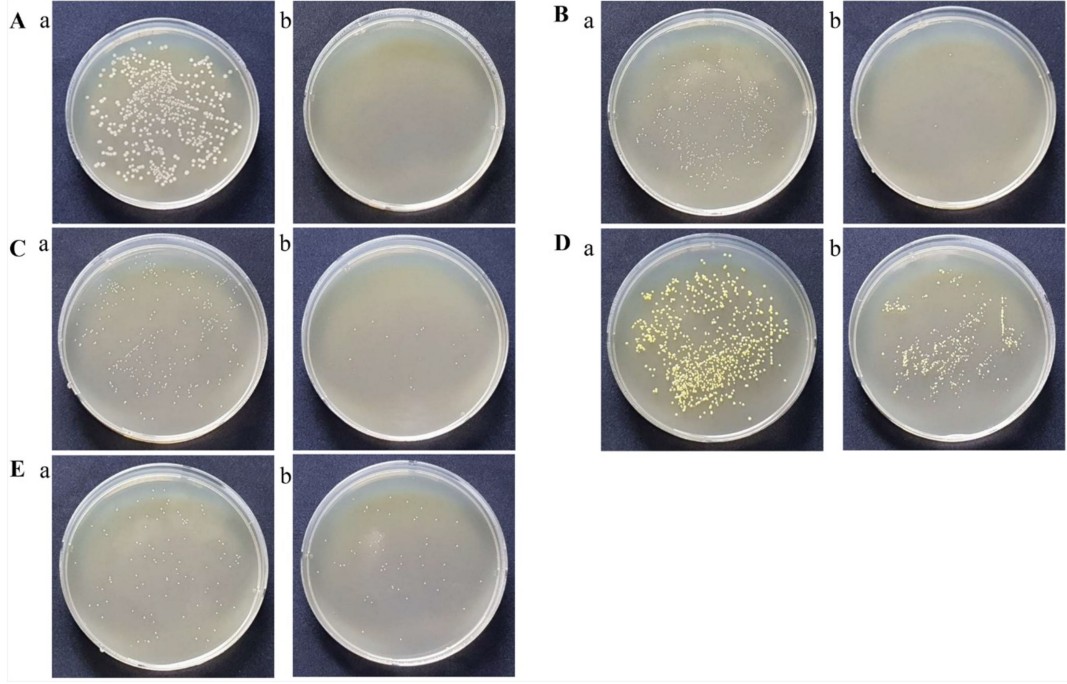

**Figure 1.** Bactericidal effect after 30 min of Puriton® treatment. The "a" figures were before Puriton® treatment, and the "b" figures were after 30 min of Puriton® treatment. Puriton® efficiently suppresses bacterial proliferation. (**A**) *Alcaligenes faecalis* ATCC1004; (**B**) *Enterococcus faecalis* ATCC29212; (**C**) *Mycobacterium smegmatis* ATCC9341; (**D**) *Micrococcus luteus* ATCC9341; (**E**) *Bacillus subtilis* ATCC6633.

### 3.2. Puriton® Suppressed the Cytopathic Effects of Zika Virus and H5N1 Virus

Neutralization controls showed that the virus was effectively detected in the titer assay. Toxicity controls showed that titer plates were valid, and no toxicity was observed on the test plates. Virucidal

results are presented in Tables 2 and 3. For Zika virus, both 4 h and 18 h of incubation with 70% EtOH were fully effective (CCID$_{50}$ of Zika virus per 100 μL was 0.7). The undiluted compound and 50% and 90% Puriton® showed effective virucidal effect after 4 h and 18 h of contact time (Table 2). The CCID$_{50}$ of Zika virus per 100 μL after treatment with 50% and 90% Puriton® was 0.7 for 4 h and 18 h of incubation. For H5N1 virus, 70% EtOH was fully effective (CCID$_{50}$ of H5N1 per 100 μL was 0.8 ± 0.2 for 4 h incubation and 0.7 for 18 h incubation). The undiluted Puriton® showed effective virucidal effects after 4 h and 18 h of contact times, whereas 50% Puriton® was more effective after 18 h of contact time than after 4 h of contact time. The CCID$_{50}$ of H5N1 virus per 100 μL after treatment with 50% Puriton® for 4 h was 1.9 ± 0.5 (Table 3).

**Table 2.** Virucidal efficacy of Puriton® against Zika virus after 4 or 18 h of liquid–liquid contact at 22 ± 2 °C.

|  | Contact Time (h) | 90% Puriton® | 50% Puriton® | 70% Puriton® | Water |
|---|---|---|---|---|---|
| [a] CCID$_{50}$ Per 100 μL | 4 | 0.7 ± 0.0 *** | 0.7 ± 0.0 *** | 0.8 ± 0.2 *** | 5.0 ± 0.00 |
| Log reduction value | 4 | >4.3 | >4.3 | 4.2 | n/a |
| [a] CCID$_{50}$ Per 100 μL | 18 | [b] <0.7 *** | 0.7 ± 0.0 *** | [b] <0.7 *** | 5.2 ± 0.3 |
| Log reduction value | 18 | >4.5 | 4.5 | >4.5 | n/a |

[a] Log10 CCID50 of virus per 0.1 mL, average ± SD of triplicate experiments; [b] for statistical analysis "<" signs were ignored; *** $p < 0.001$ compared with negative control (water), analyzed by one-way ANOVA and Dunnett post-test.

**Table 3.** Virucidal efficacy of Puriton® against Influenza A virus (H5N1) after 4 or 18 h of liquid–liquid contact at 22 ± 2 °C.

|  | Contact Time (h) | 90% Puriton® | 50% Puriton® | 70% Puriton® | Water |
|---|---|---|---|---|---|
| [a] CCID$_{50}$Per 100 μL | 4 | [b] <0.7 *** | 1.9 ± 0.5 *** | [b] <0.7 *** | 4.5 ± 0.2 |
| Log reduction value | 4 | >3.8 | 2.6 | >3.8 | n/a |
| [a] CCID$_{50}$Per 100 μL | 18 | [b] <0.7 *** | <0.7 *** | [b] <0.7 *** | 4.1 ± 0.4 |
| Log reduction value | 18 | >3.4 | >3.4 | >3.4 | n/a |

[a] Log$_{10}$ CCID$_{50}$ of virus per 0.1 mL, average ± SD of triplicate experiments; [b] for statistical analysis "<" signs were ignored; *** $p < 0.001$ compared with negative control (water), analyzed by one-way ANOVA and Dunnett post-test.

### 3.3. Puriton® Oral Administration Was Safe at under 800 μL

To evaluate the safety of Puriton®, a 26-week toxicity testing of repeated oral administration of Puriton® was conducted. The maximum dose was set as 800 μL, considering the maximum dose administered to one mouse (approximately 1000 μL). During the experiment, several safety-related factors were observed every day, namely locomotion and behavior, but body weight (Figure 2A) and food intake (or consumption, Figure 2B) were measured over several-day intervals. There was no significant change in body weight between groups. Puriton® slightly increased body weight in a dose-dependent manner, but the increase was not statistically significant. After treatment, slight differences in body weight were observed between the 800 μL Puriton® group and the other groups.

There was no difference in population of white blood cells (WBCs) between the control, 400 μL Puriton®, and 800 μL Puriton® groups (Figure 3A and Supplementary Materials). Puriton® increased the number of neutrophils in a dose-dependent manner (Figure 3B and Supplementary Materials), but did not alter the number of other WBCs (Figure 3C,E and Supplementary Materials).

There was no difference in total cholesterol level among the groups (Figure 4A and Supplementary Materials). LDL levels were the lowest in the 400 μL Puriton® group, and there was no difference in LDL levels between the control and 800 μL Puriton® groups (Figure 4B and Supplementary Materials). HDL levels in the 800 μL Puriton® group were higher than that in the 400 μL Puriton® group (Figure 4C and Supplementary Materials). Puriton® treatment effectively suppressed blood glucose level (Figure 4D and Supplementary Materials).

To evaluate the toxicity of Puriton® on five important organs, namely the brain, heart, kidney, liver, and lungs, the organs were observed with hematoxylin and eosin staining. However, the results showed that Puriton® caused no histopathological change in these organs (Figure 5).

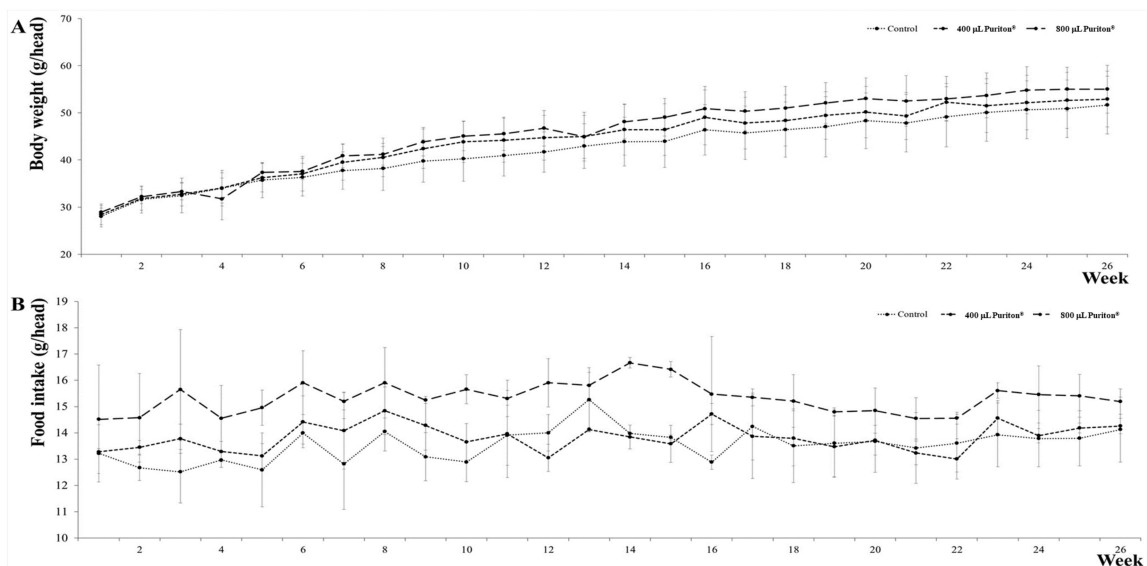

**Figure 2.** Changes in body weight and food intake per week of 26-week treatment. The dots represent the mean and SD. (**A**) Body weight changes during the study; (**B**) Food intake changes during the study.

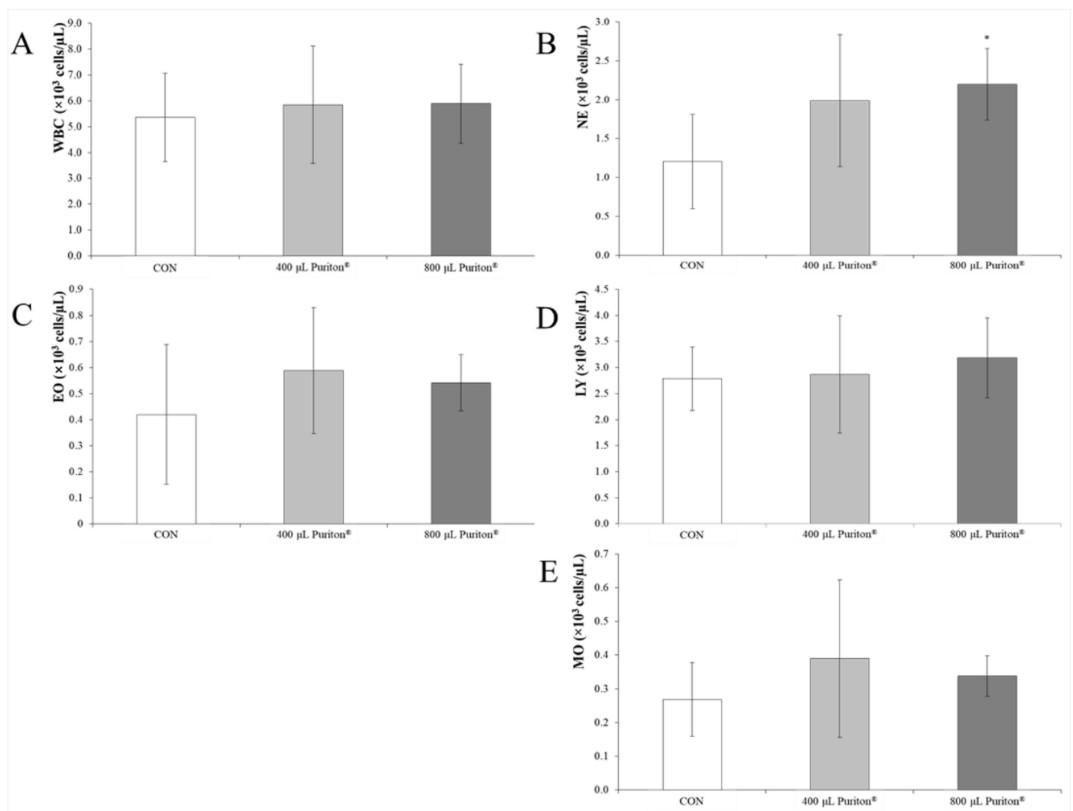

**Figure 3.** Results of white blood cell and differential cell counts. Puriton® slightly increased the number of neutrophils. (**A**), the difference of WBCs among the groups; (**B**), the difference of neutrophils among the groups; (**C**), the difference of eosinophils among the groups; (**D**), the difference of lymphocytes among the group; (**E**) the difference of monocyte among the groups. WBC, white blood cell; NE, neutrophil; EO, eosinophil; LY, lymphocyte; MO, monocyte. Each bar represents the mean ± SD (n = 6). * $p < 0.05$ vs. control (CON) group.

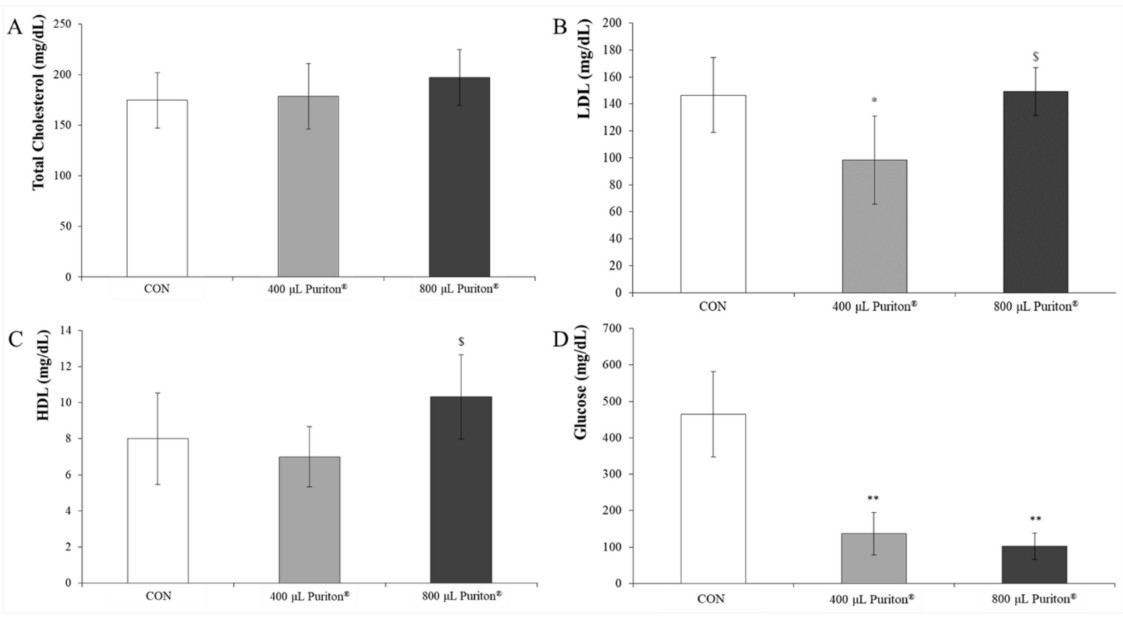

**Figure 4.** Results of blood chemistry analysis. Although LDL level was the highest, the HDL level was much higher than the level of other parameters. Puriton® significantly decreased glucose levels. (**A**) Total cholesterol; (**B**) LDL; (**C**) HDL; (**D**) Glucose. LDL, low density lipid; HDL, high density lipid. Each bar represents the mean ± SD (n = 6). * $p < 0.05$ vs. CON group; ** $p < 0.001$ vs. CON group; $ $p < 0.05$ vs. The 400 μL Puriton® group.

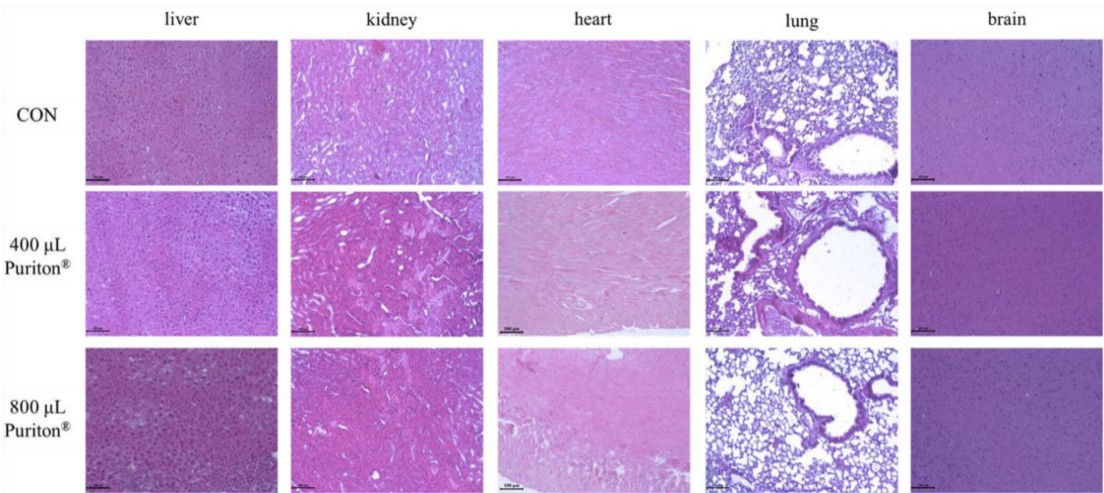

**Figure 5.** Results of histopathological analysis of Puriton® in the brain, heart, kidney, liver, and lungs. Scale bar, 100 μm; magnification, ×200.

## 4. Discussion

Puriton® is a solution of mineral mixture consisting of 20% biotite, 17% kaolinite, 16% montmorillonite, 12% serpentine, 8% mica, and other minerals in saline solution. Anti-bacterial effects of biotite had been reported, such that it could remove *Salmonella enterica* serovar via Th1-specific responses [26]. Kaolinite is 16% of Puriton® and is widespread in the world. It has several biological efficacies, such as antibacterial effects, anti-viral effects, and detoxification [27]. Qin et al. reported that $Fe^{3+}$-saturated montmorillonite could suppress bacterial activity in municipal wastewater effluents [28]. Germanium biotite has been used as a dietary material for preventing porcine reproductive and respiratory syndrome viruses [29]. Mica stimulates the activation of the immune system via activation of macrophages, lysosomes, and phagosomes [30], and can inhibit virus activation. Puriton® prevented the proliferation

of nine pathogenic bacteria up to 100% and 91.1% in *B. subtilis* ATCC6633, and inactivated not only Zika virus ($CCID_{50}$ per 100 μL was 0.7 ± 0.0 at 4 and 18 h) but also the H5N1 virus ($CCID_{50}$ per 100 μL was 1.9 ± 0.5 at 4 h and <0.7 at 18 h). From the results of the 26-week toxicity testing of repeated oral administration of Puriton®, although Puriton® could influence the commensal flora of the gastrointestinal tract, we found that Puriton® treatment was not only safe, but exerted therapeutic effects, such as an immune-modulating effect (increasing the neutrophil population, Figure 3B), an LDL-suppressing effect by increasing HDL levels (Figure 4B,C), and blood glucose-controlling effects (Figure 4D).

In addition, metabolic syndromes, such as diabetes, dyslipidemia, hypertension, inflammation, and hyperglycemia, are closely correlated with aging [31]. Dyslipidemia indicates increased LDL levels and decreased HDL levels in blood, and it may ultimately induce blood-flow dysfunction [32]. However, oral Puriton® administration exerted a dyslipidemia-modulating effect; it not only decreased LDL levels at 400 μL, but also increased HDL levels at 800 μL. Considering that the important point of dyslipidemia modulation is lower levels of LDL than HDL, Puriton® might be a candidate for effective dyslipidemia modulating. Diabetes mellitus is characterized by hyperglycemia, and Puriton® significantly decreased blood glucose level from 464.0 to 116.90 mg/dL in the control group to 101.5 ± 36.40 mg/dL in the 800 μL Puriton® group. Therefore, Puriton® may be a candidate modulator of diabetes mellitus.

Bacterial infection could lead to severe diseases of the respiratory and intestinal system; in fact, lower respiratory infection was one of the top 10 causes of death worldwide in 2016 [1]. Although synthetic antibiotics have been developed to control microorganisms, these antibiotics exhibit many adverse effects. For example, telithromycin causes visual disturbances, taste and smell disorders, severe liver damage, and loss of consciousness [33]. Moreover, hepatic side-effects [34], destruction of normal flora [35], and hazards to male fertility [36] were reported.

## 5. Conclusions

Considering that Puriton® exerted inhibitory effects not only on the proliferation of nine pathogenic bacteria but also on the replication of the Zika virus and H5N1 virus, we concluded that Puriton® might a candidate for functional food additives, cosmetics, and drugs with antimicrobial effect. However, in further studies the mechanisms of bactericidal and virucidal effects of Puriton® should be elucidated.

**Supplementary Materials:** The following are available online at http://www.mdpi.com/2227-9717/8/11/1481/s1, S1 Dataset: Raw data for Figures 3 and 4.

**Author Contributions:** Conceptualization, K.-H.K. and D.-H.P.; Data curation, S.-H.B., M.-H.K., M.-J.L. and C.-S.B.; Formal analysis, S.-H.B. and S.-Y.L.; Funding acquisition, K.-H.K.; Investigation, S.-H.B., M.-H.K., S.-Y.L. and M.-J.L.; Methodology, S.-H.B., M.-H.K., S.-Y.L., C.-S.B. and M.-J.L.; Project administration, K.-H.K. and D.-H.P.; Resources, K.-H.K.; Supervision, K.-H.K. and D.-H.P.; Visualization, K.-H.K. and D.-H.P.; Writing—original draft, S.-H.B.; Writing—review and editing, K.-H.K. and D.-H.P. All authors have read and agreed to the published version of the manuscript.

**Funding:** This research was conducted with financial support by Kadesh, Inc.

**Conflicts of Interest:** The authors declare that there is no conflict of interest.

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
