# Peer review of "Bactericidal and Virucidal Efficacies and Safety of Puriton®"

_processes, doi:10.3390/pr8111481_

Round 1
Reviewer 1 Report
The authors describe an investigation of the antimicrobial activities and in vivo safety of Puriton, a mixture of nature minerals and clays. The mixture was tested in its efficacy in vitro to control several medically-important bacteria and viruses, and also in vivo in mice for several safety measures.
The authors present data showing that Puriton shows broad antimicrobial activity, especially against Gram negative bacteria. They test the mixture by oral administration in mice for safety measures, and find no differences from the control for most, except for a statistically significant reduction in blood glucose.
The data is carefully presented and the statistical analysis appears to be sufficient. The study is an important addition to the literature on antimicrobial clays/minerals. It is overall very carefully and clearly written and the Figures/Tables nicely presented, but could be improved by one proofreading by a native speaker, if possible.
Major points
- I would also especially like to see something included about the safety of orally-administered Puriton regarding the gastrointestinal tract.
- I believe that the amount of background in the Introduction and the placement of the study within the context of other clay minerals is lacking.
- Several details about bacterial growth, where the bacteria were obtained, are missing
- I am not sure the histopathology (Fig 5) is convincing (esp for the heart). Were these samples assessed blindly?
Minor points
Abstract
- It is not clear which animal model is being used. Also, it should be made clear that two separate investigations were performed: 1) antimicrobial activity in vitro and 2) safety in vivo
Introduction
- Heme, for example, is given as an example of a mineral. For the purposes of this paper, I would only talk about true geological minerals/clays.
Methods
- Please describe how the bacteria were grown. Which medium?
- How was the dose of Puriton selected?
Table 1.
- How was the chemical composition of Puriton obtained? Reference?
Table 2.
- What was the limit of detection? For example, if it was 100 CFU/ml, please report “0” as <100
- check spelling of “Escherichia”
- Line 122 - since only a single time point was assayed, no conclusion about the death “rate” can be made
Discussion
- Line 196: In Table 1, the authors present that Puriton is a mixture of several iron-rich minerals. There have been several studies that report that clays with certain ions might underlie their antimicrobial activity. Therefore, the authors should discuss this, and if this might be the mechanism of action for Puriton
- Can the authors comment on the different efficacy of Puriton in the different bacteria?
- Lines 207 - 214 - the connection with lipids is not very strong, statistically. The effect on glucose is interesting, but not strong. Can the authors discuss what might be the mechanism for this? Has a similar response been seen before after administration of other clays/minerals?
Author Response
Major points
- I would also especially like to see something included about the safety of orally-administered Puriton regarding the gastrointestinal tract.
Ans) Thank you so much for your informative concern of Puriton’s toxicity. However, that has been approved as a water mineral supplement by Korean Food and Drug Agency. In this study we hoped that we confirmed the safety of Puriton.
- I believe that the amount of background in the Introduction and the placement of the study within the context of other clay minerals is lacking.
Ans) Thank you so much for the informative comments and I added the usage of clay mineral.
- Several details about bacterial growth, where the bacteria were obtained, are missing
Ans) Thank you so much for the generous comments and I add the sentences about the information of obtain site.
- I am not sure the histopathology (Fig 5) is convincing (esp for the heart). Were these samples assessed blindly?
Ans) Thank you so much for the comments and these samples were blindly assessed. However, the photos of heart could make you and the readers to be confused and then I changes the photos about the heart.
Minor points
Abstract
- It is not clear which animal model is being used. Also, it should be made clear that two separate investigations were performed: 1) antimicrobial activity in vitro and 2) safety in vivo
Ans) Thank you so much for the comments and I amended the sentence.
Introduction
- Heme, for example, is given as an example of a mineral. For the purposes of this paper, I would only talk about true geological minerals/clays.
Ans) Thank you so much for the informative comments and I added the usage of clay mineral.
Methods
- Please describe how the bacteria were grown. Which medium?
Ans) Thank you so much for the comments and I added the sentences.
- How was the dose of Puriton selected?
Ans) I added the sentences and in order to evaluate the baceriocidal effect of Puriton we used distilled water or Puriton as a medium. And then we didn’t decide the dose of Puriton.
Table 1.
- How was the chemical composition of Puriton obtained? Reference?
Ans) Puriton was made of this composition, Table 1.
Table 2.
- What was the limit of detection? For example, if it was 100 CFU/ml, please report “0” as <100
Ans) Thank you so much for the comments and I added the sentences.
- check spelling of “Escherichia”
Ans) Thank you so much for the generous comment and I amended the spelling.
- Line 122 - since only a single time point was assayed, no conclusion about the death “rate” can be made
Ans) Thank you so much for your comment but we calculated the inhibition rate at 30 minute after treatment based on the initiation point, 0 minute.
Discussion
- Line 196: In Table 1, the authors present that Puriton is a mixture of several iron-rich minerals. There have been several studies that report that clays with certain ions might underlie their antimicrobial activity. Therefore, the authors should discuss this, and if this might be the mechanism of action for Puriton
Ans) Thank you so much for the generous comments and I added several sentences with references.
- Can the authors comment on the different efficacy of Puriton in the different bacteria?
Ans) Thank you so much but now I can’t answer the question. I think we need the further study.
- Lines 207 - 214 - the connection with lipids is not very strong, statistically. The effect on glucose is interesting, but not strong. Can the authors discuss what might be the mechanism for this? Has a similar response been seen before after administration of other clays/minerals?
Ans) Thank you so much but now I can’t answer the question. I think we need the further study.
Reviewer 2 Report
General comment: The manuscript „ Bactericidal and Virucidal Efficacies and Safety of Puriton®” by So-Hyeon Bokand co-workers investigated the antibacterial and antiviral effects of a new mineral mixture called Puriton®. In the addition to the evaluation of the antimicrobial characteristics of Puriton®, the author also investigated the safety of this product via an oral treatment period of 26-weeks in mice. Due to the antibacterial and virus inactivating properties of Puriton® the author concluded that this mineral mixture has the potential as a functional food additive, when applied in concentrations of lower 800µl. The manuscript is suitable for publication in Processes, although some major points need to be addressed:
Major comments:
- Introduction
- Please provide more information about relevant diseases, such as diabetes and others and try to build a link between the needed information in the introduction and the need of this information in the discussion part.
- L61-62 mention that “the antimicroorganism effect of minerals is unknown.” This sentences does not seem appropriate as authors like Moosavi (2017), Haydel et al. (2008), Williams & Haydel (2010; DOI: 10.1080/00206811003679737) and Williams (2019; DOI: 10.1007/s42860-018-0002-8) have already described the antibacterial potential of different minerals and mineral clays. Please add some more information about the antibacterial effects and virus absorbing effects of minerals/mineral clays as well as their mechanisms of action on bacteria and viruses.
- L62-63 the concrete goals of the study are underrepresented and scarcely described. Please add some more information about the aims of this study, the used methods and also why an oral treatment study was applied.
- Materials and Methods
- Please clarify how Puriton® was applied (liquid, solid,…)? What kind of carrier material was used for the application of Puriton® (aqueous solution, emulsion,…). Please provide more concrete product data to increase transparency. I highly recommend to mention some physical parameters such as pH and conductivity of the used Puriton® as well.
- Table 1 exhibits the percentage distribution of each mineral component, whereas the author has not clearly described whether this distribution is related to the total mineral content of this product or if the distribution is compared to the total product content, please clarify. Moreover the addition of absolute content values (w/v, w/w,…) of each mineral would also allow a direct comparability to other studies.
- „2.1. Bacterial Culture and Viability Examination.“ Please add some information about the used bacterial amount and the used amount/concentration of Puriton ®. Do the authors washed the bacterial cells after 30 min incubation or was the Puriton® also transferred to the plate?
- L68-69 after 30 min incubation the number of bacterial colonies was counted? Does the author use a flow cytometer? If not please also add the incubation time of the inoculated plates and the respective incubation temperature of each bacterial strain.
- Results
- Please add the standard deviation of each bacterial strain to the table 2.
- L154-155 please describe why an oral treatment study was performed. As mentioned above, the reader is kind of lost why the author decided to test the safety of this product via an oral treatment study and not for example via a skin toxicity test. If there are other experimental data available about this product I suggest to provide them (introduction or discussion), that would make some points clearer.
- Discussion
- L199-203 Does the author also investigated the infection marker of the mice to exclude a slight inflammation, which could be also a cause for increased neutrophil population.
- Please add more information abut possible mechanisms of action of Puriton® on viruses. Could be the pH value of the Puriton® affect the virus particle infectivity, as it is known that viruses can be inactivated due to high and low pH values. Moreover as the author is using minerals with different charges, I also suggest to add some information about the influence of different charges on the viral particles used in this study. Please provide some proofed data.
- Has the author proofed whether the viruses are more likely to bind on the charged minerals than on the host cells. If yes, could it be possible that the majority of virus particle was removed with the first cell wash without ever having infected the cells (control of the exchanged supernatant after 4h). Please provide some proofed data.
- Please also ad some detailed information about the mechanisms of action of minerals to bacteria. How does the author explain the susceptibility differences seen between gram-positive and gram-negative bacterial strains?
- Conclusion
- At the moment the conclusion is too imprecise. It is not clear whether the study could answer the questions that should be outlined in the aim of this study.
Minor comments:
- L25, 27, 28, 29, 31, 63, 122, 125, 127,128, 136, 137,… “Puriton” should be changed to “Puriton®” in the whole manuscript.
- L39-40 “…the common death causes was lower respiratory infections existed.“ should be changed to “…the most common death causes worldwide was lower respiratory infections.
- L40-41 The sentence before mentioned that lower respiratory infections are the major death causes worldwide and in L40-41 the author went on with the listing of different disease, which are not associated with lower respiratory infections (except of TBC). The addition of one or two more sentences could be helpful to mention that also diarrhoeal diseases are a problem worldwide.
- L198 “eliminated” should be changed to “inactivated”
- L207-209 and L214-218 I suggest to describe these diseases, their relevance and their relevance within this study in the introduction. These information would also indicate why the safety of Puriton® in oral treatment study was determined.
- L220-225 I suggest to put this part also into the introduction, as no link between these diseases and Puriton® was made by the author.
Author Response
Major comments:
- Introduction
- Please provide more information about relevant diseases, such as diabetes and others and try to build a link between the needed information in the introduction and the need of this information in the discussion part.
Ans) Thank you so much for the informative comments and I added the sentences in the introduction section.
- L61-62 mention that “the antimicroorganism effect of minerals is unknown.” This sentences does not seem appropriate as authors like Moosavi (2017), Haydel et al. (2008), Williams & Haydel (2010; DOI: 10.1080/00206811003679737) and Williams (2019; DOI: 10.1007/s42860-018-0002-8) have already described the antibacterial potential of different minerals and mineral clays. Please add some more information about the antibacterial effects and virus absorbing effects of minerals/mineral clays as well as their mechanisms of action on bacteria and viruses.
Ans) Thank you so much for the comments and I removed the sentences.
- L62-63 the concrete goals of the study are underrepresented and scarcely described. Please add some more information about the aims of this study, the used methods and also why an oral treatment study was applied.
Ans) Thank you so much and I amended the sentences.
2. Materials and Methods
- Please clarify how Puriton® was applied (liquid, solid,…)? What kind of carrier material was used for the application of Puriton® (aqueous solution, emulsion,…). Please provide more concrete product data to increase transparency. I highly recommend to mention some physical parameters such as pH and conductivity of the used Puriton® as well.
Ans) Thank you so much and I added the sentences.
- Table 1 exhibits the percentage distribution of each mineral component, whereas the author has not clearly described whether this distribution is related to the total mineral content of this product or if the distribution is compared to the total product content, please clarify. Moreover the addition of absolute content values (w/v, w/w,…) of each mineral would also allow a direct comparability to other studies.
Ans) Thank you so much and I added the sentences.
- „2.1. Bacterial Culture and Viability Examination.“ Please add some information about the used bacterial amount and the used amount/concentration of Puriton®. Do the authors washed the bacterial cells after 30 min incubation or was the Puriton® also transferred to the plate?
Ans) Thank you so much and I added the sentences.
- L68-69 after 30 min incubation the number of bacterial colonies was counted? Does the author use a flow cytometer? If not please also add the incubation time of the inoculated plates and the respective incubation temperature of each bacterial strain.
Ans) Thank you so much and I added the sentences.
3. Results
- Please add the standard deviation of each bacterial strain to the table 2.
Ans) Thank you so much for the comment but we had one time experiment as the results were clearly.
- L154-155 please describe why an oral treatment study was performed. As mentioned above, the reader is kind of lost why the author decided to test the safety of this product via an oral treatment study and not for example via a skin toxicity test. If there are other experimental data available about this product I suggest to provide them (introduction or discussion), that would make some points clearer.
Ans) Thank you so much for the informative comments and I added the explanation.
4. Discussion
- L199-203 Does the author also investigated the infection marker of the mice to exclude a slight inflammation, which could be also a cause for increased neutrophil population.
Ans) Thank you so much for the informative comments and in the next study I will investigate the inflammation effect of Puriton®.
- Please add more information abut possible mechanisms of action of Puriton® on viruses. Could be the pH value of the Puriton® affect the virus particle infectivity, as it is known that viruses can be inactivated due to high and low pH values. Moreover as the author is using minerals with different charges, I also suggest to add some information about the influence of different charges on the viral particles used in this study. Please provide some proofed data.
Ans) Thank you so much for the informative comments and I added several sentences. However, now it is not easy to provide proofed data as we should have another study, In the further study we will find important reasons depending on the comments.
- Has the author proofed whether the viruses are more likely to bind on the charged minerals than on the host cells. If yes, could it be possible that the majority of virus particle was removed with the first cell wash without ever having infected the cells (control of the exchanged supernatant after 4h). Please provide some proofed data.
Ans) Thank you so much for the informative comments but please let me know this situation. Now it is not easy to provide proofed data as we should have another study, In the further study we will find important reasons depending on the comments.
- Please also ad some detailed information about the mechanisms of action of minerals to bacteria. How does the author explain the susceptibility differences seen between gram-positive and gram-negative bacterial strains?
Ans) Thank you so much for the informative comments and according to the results about preventing the proliferation of nine bacteria Puriton® can suppress most bacteria regardless of gram-positive and gram-negative bacteria.
5. Conclusion
- At the moment the conclusion is too imprecise. It is not clear whether the study could answer the questions that should be outlined in the aim of this study.
Ans) Thank you so much for the informative comments and I amended the sentences.
Minor comments:
- L25, 27, 28, 29, 31, 63, 122, 125, 127,128, 136, 137,… “Puriton®” should be changed to “Puriton®” in the whole manuscript.
Ans) I amended them.
- L39-40 “…the common death causes was lower respiratory infections existed.“ should be changed to “…the most common death causes worldwide was lower respiratory infections.
Ans) Thank you so much for the generous comment and I amended the sentence.
- L40-41 The sentence before mentioned that lower respiratory infections are the major death causes worldwide and in L40-41 the author went on with the listing of different disease, which are not associated with lower respiratory infections (except of TBC). The addition of one or two more sentences could be helpful to mention that also diarrhoeal diseases are a problem worldwide.
Ans) Thank you so much for the generous comments. I amended the sentence and added sentence.
- L198 “eliminated” should be changed to “inactivated”
Ans) Thank you so much for the generous comment and I amended the word.
- L207-209 and L214-218 I suggest to describe these diseases, their relevance and their relevance within this study in the introduction. These information would also indicate why the safety of Puriton® in oral treatment study was determined.
Ans) Thank you so much for the generous and informative comments but I can’t find the sentences, L207-209 and L214-218. I think it is different between the submitted manuscript and the manuscript for peer-reviewing. Please let me know the accurate sentences and then if I have chance to revise I willing to change them.
- L220-225 I suggest to put this part also into the introduction, as no link between these diseases and Puriton® was made by the author.
Ans) Thank you so much for the generous and informative comments but I can’t find the sentences, L220-225. I think it is different between the submitted manuscript and the manuscript for peer-reviewing. Please let me know the accurate sentences and then if I have chance to revise I willing to change them.
Reviewer 3 Report
Dear authors!
The article „Bactericidal and Virucidal Efficacies and Safety of 2 Puriton®” touches on an interesting topic of the use of an antibacterial formulation, Puriton.
I think the article describes an interesting phenomenon with a potential for innovation. However, the manuscript itself leave much to be desired and I must regretfully conclude that it is unfit for publication in Processes its current state. The structure of the article is underdeveloped, at times I am even under impression it is in fact a random pool of haphazardly assembled experiments. The presentation of the results also requires attention, some figures are illegible or underdeveloped. I hope that the following suggestions will be of use for the authors in future work on this project.
Suggested corrections:
20 mice???? Are you sure? Or mica ?
22-23 suggests that Escherichia coli is always pathogenic, likewise for Micrococcus luteus, Bacillus subtilis. This should be rewritten.
25 „effectively controlled the bacterial population”. That could be misleading. Clarification is required.
30 I have doubts regarding long term safety of the use of Puriton, among other factors because of the presence of serpentine. In the case of minerals capable of accumulation in living organisms, including serpentine, which could naturally contain small amounts of among others chrysotile asbestos, there is a risk of cancerogenic effects in the case of prolonged use. The key factor here is the long term effect, which was not investigated.
The Abstract is poor. It should be rewritten from the scratch. There are many language errors in it, for example:
18-19 Either top or leading, not both
20-21 is a jumble, perhaps: … and its bactericidal efficiency was evaluated?
30-31 “…and based on histopathological changes in the brain, heart, kidney, liver, and lung…”
Strongly suggests there were such changes. It has to be clarified.
The linguistic side of the work is generally poor and would have benefited from a correction by a native speaker.
Other than that, there are several other issues:
36 and 18 look like a copy, please edit it.
42-43 Does E. coli always do that?
45-46 S. aureus is an opportunistic pathogen and has to be treated as such, especially in the context of its occurrence on the surface of the skin.
65 Under the header Bacterial Culture and Viability Examination
Lack of description of details.
What medium was used?
There is no information regarding what collection were the bacterial strains sourced from. This is unacceptable.
The contents of the formulation should have been described under its own header.
116 The paragraph name is excessively suggestive. It does not fit a scientific work.
117 Information regarding the strains has to be moved to Materials. It has no place in Results.
Table 2 has to be restructured
Fig 1 The columns should be named, as in the description, but shortened. It would have made it easier to read.
Fig. 3 It is unfit for publication. It is unclear.
The Discussion is poorly written and very brief. It does not exhaust the material provided by the Results by far.
Tests of potential mutagenic effects are also missing.
Author Response
Suggested corrections:
20 mice???? Are you sure? Or mica ?
Ans) Thank you and I amended the word.
22-23 suggests that Escherichia coli is always pathogenic, likewise for Micrococcus luteus, Bacillus subtilis. This should be rewritten.
Ans) Thank you so much for the comment but I can’t understand that. Please let me know what the reviewer meant.
25 „effectively controlled the bacterial population”. That could be misleading. Clarification is required.
Ans) I amended the sentence.
30 I have doubts regarding long term safety of the use of Puriton®, among other factors because of the presence of serpentine. In the case of minerals capable of accumulation in living organisms, including serpentine, which could naturally contain small amounts of among others chrysotile asbestos, there is a risk of cancerogenic effects in the case of prolonged use. The key factor here is the long term effect, which was not investigated.
Ans) Thank you but under 26-week oral repeated study we can’t find any toxicity.
The Abstract is poor. It should be rewritten from the scratch. There are many language errors in it, for example:
18-19 Either top or leading, not both
Ans) Thank you and I amended the expression.
20-21 is a jumble, perhaps: … and its bactericidal efficiency was evaluated?
Ans) I amended the sentence.
30-31 “…and based on histopathological changes in the brain, heart, kidney, liver, and lung…”
Ans) I amended the sentence.
Strongly suggests there were such changes. It has to be clarified.
The linguistic side of the work is generally poor and would have benefited from a correction by a native speaker.
Ans) The manuscript was edited by editing company but I willing to send the manuscript to editing company again.
Other than that, there are several other issues:
36 and 18 look like a copy, please edit it.
Ans) I amended the sentence, L18.
42-43 Does E. coli always do that?
Ans) E. coli is one of bacteria which cause foodborne disease. Although they exist in the large intestine they are normal flora when they are in the food they can induce the foodborne disease.
45-46 S. aureus is an opportunistic pathogen and has to be treated as such, especially in the context of its occurrence on the surface of the skin.
Ans) I described that in L46-47.
65 Under the header Bacterial Culture and Viability Examination
Ans) Thank you so much for the generous comment.
Lack of description of details.
What medium was used?
Ans) I added the sentences.
There is no information regarding what collection were the bacterial strains sourced from. This is unacceptable.
Ans) I added the sentences.
The contents of the formulation should have been described under its own header.
Ans) I added the sentences.
116 The paragraph name is excessively suggestive. It does not fit a scientific work.
Ans) I added the sentences.
117 Information regarding the strains has to be moved to Materials. It has no place in Results.
Ans) In this paragraph I described the results of bactericidal effect of Puriton®.
Table 2 has to be restructured
Ans) Thank you so much for the comment. However, I think that Table 2 is very clear and then it is easy that the readers understand the result.
Fig 1 The columns should be named, as in the description, but shortened. It would have made it easier to read.
Ans) I added the sentences.
Fig. 3 It is unfit for publication. It is unclear.
Ans) These results are basic data for evaluating specific agent’s safety.
The Discussion is poorly written and very brief. It does not exhaust the material provided by the Results by far.
Ans) Thank you so much for the comments. And I amended the Discussion section.
Tests of potential mutagenic effects are also missing.
Ans) Thank you so much for the comment but in the further study we should confirm the mutagenesis by Puriton®.
Round 2
Reviewer 2 Report
Major comments:
- Discussion
- The author responded that the inflammation effect of Puriton® will be investigated in a following study. Based on this answer as well as the author has not provided information about the inflammation potential of clays via oral uptake in general, the statement “…that Puriton® treatment was not only safe, but exerted therapeutic effects, such as immune-modulating effect (increasing the neutrophil population, …” seems not appropriate as other reasons for such an increase could not be fully excluded within this study. I suggest removing the statement about the immune modulating effect. If the author can proof that similar effects has been seen in other studies under similar conditions, this would support the statement made and it could be included again in the discussion.
- L230-231 the author mentioned that Puriton® may enhance the innate immunity. The author should also mention that an increased number of neutrophils could also be an indicator for inflammation processes.
Minor comments:
- Introduction
- L55-56 the transmission from bacterial disease causes to diabetes could be smoothed out by the addition of another sentence for example “Beside bacterial infections also an increase in metabolic disorders, like diabetes, could be observed worldwide in the last few years”.
- L65 Please remove “et al”.
- Results
- Please mention that the experiments shown in table 2 have performed single on one day, to prevent misunderstandings.
Author Response
Discussion
The author responded that the inflammation effect of Puriton® will be investigated in a following study. Based on this answer as well as the author has not provided information about the inflammation potential of clays via oral uptake in general, the statement “…that Puriton® treatment was not only safe, but exerted therapeutic effects, such as immune-modulating effect (increasing the neutrophil population, …” seems not appropriate as other reasons for such an increase could not be fully excluded within this study. I suggest removing the statement about the immune modulating effect. If the author can proof that similar effects has been seen in other studies under similar conditions, this would support the statement made and it could be included again in the discussion.
L230-231 the author mentioned that Puriton® may enhance the innate immunity. The author should also mention that an increased number of neutrophils could also be an indicator for inflammation processes.
Ans) Thank you so much for the specific comments and based on the comments I deleted the sentence.
Minor comments:
Introduction
L55-56 the transmission from bacterial disease causes to diabetes could be smoothed out by the addition of another sentence for example “Beside bacterial infections also an increase in metabolic disorders, like diabetes, could be observed worldwide in the last few years”.
Ans) Thank you so much for the generous comments and I amended the sentence using with suggested comment.
L65 Please remove “et al”.
Ans) Thank you so much and I deleted the words.
Results
Please mention that the experiments shown in table 2 have performed single on one day, to prevent misunderstandings.
Ans) Thank you so much for the generous comment and I added the sentence.
Reviewer 3 Report
It suggests the following fixes:
Abstract
22 Is „pathogenic bacteria” I suggest „pathogens and opportunistic bacteria”
Introduce
Introduce in my opinion is better but still required improve style and poor language in microbiology aspects.
56-60 This part is „alone island” in this introduce. It is required better include it to this text.
65: Too colloquial language „Clay can catch the toxin”
Materials and Methods
75: „and Viability Examination” may „and culturable conditions”
86: I recomend include heading „Materials”. Move text above „Bacterial Culture”
Shortly describe the origin of the product, not only the company. Place the table 1 in the text.
Results
The headline titles need to be generalized and shortened. Should not be suggestive like "effectively eradicated the colonies of nine pathogenic bacteria."
135-136 „eradicate” is to much/ not clear - it should be replacement on ect." strongly inhibition growth"
Fig 1 The columns should be named, as in the description, but shortened. It would have made it easier to read.
Discussion
The discussion does not emphasize the possible influence of the preparation on the commensal flora of the gastrointestinal tract.
Author Response
Abstract
22 Is „pathogenic bacteria” I suggest „pathogens and opportunistic bacteria”
Ans) Thank you so much for the generous comment and based on the comment I changed the words.
Introduce
Introduce in my opinion is better but still required improve style and poor language in microbiology aspects.
56-60 This part is „alone island” in this introduce. It is required better include it to this text.
Ans) Thank you so much for the generous comment and based on the comment I changed the sentence.
65: Too colloquial language „Clay can catch the toxin”
Ans) Thank you so much for the generous comment and based on the comment I changed the words.
Materials and Methods
75: „and Viability Examination” may „and culturable conditions”
Ans) Thank you so much for the generous comment and I added the words.
86: I recomend include heading „Materials”. Move text above „Bacterial Culture”
Shortly describe the origin of the product, not only the company. Place the table 1 in the text.
Ans) Thank you so much for the generous comment and based on the comment I changed the paragraphs.
Results
The headline titles need to be generalized and shortened. Should not be suggestive like "effectively eradicated the colonies of nine pathogenic bacteria."
Ans) Thank you so much for the generous comment and based on the comment I changed the headline titles.
135-136 „eradicate” is to much/ not clear - it should be replacement on ect." strongly inhibition growth"
Ans) Thank you so much for the generous comment and based on the comment I changed the sentence.
Fig 1 The columns should be named, as in the description, but shortened. It would have made it easier to read.
Ans) Thank you so much for the generous comment and based on the comment I changed the paragraphs.
Discussion
The discussion does not emphasize the possible influence of the preparation on the commensal flora of the gastrointestinal tract.
Ans) Thank you so much for the generous comment and I added the sentence.